# Association between Type 1 Diabetes Mellitus and Periodontal Diseases

**DOI:** 10.3390/jcm12031147

**Published:** 2023-02-01

**Authors:** Rosana Costa, Blanca Ríos-Carrasco, Luís Monteiro, Paula López-Jarana, Filipa Carneiro, Marta Relvas

**Affiliations:** 1Department of Medicine and Oral Surgery, University Institute of Health Sciences (IUCS-CESPU), 4585-116 Gandra, Portugal; 2Department of Periodontology, School of Dentistry, Universidad de Sevilla, C/Avicena S/n, 41009 Sevilla, Spain; 3Oral Pathology and Rehabilitation Research Unit (UNIPRO), University Institute of Health Sciences (IUCS-CESPU), 4585-116 Gandra, Portugal; 4Hospitalar Center of Tâmega e Sousa, Avenida do Hospital Padre Américo 210, 4560-136 Penafiel, Portugal

**Keywords:** periodontitis, type 1 diabetes *mellitus*, glycemic control, pro-inflammatory cytokines

## Abstract

Gingivitis and periodontitis are chronic inflammatory diseases that affect the supporting tissues of the teeth. Although induced by the presence of bacterial biofilms, other factor, such as tobacco smoking, drugs, and various systemic diseases, are known to influence their pathogenesis. Diabetes *mellitus* and periodontal diseases correspond to inflammatory diseases that have pathogenic mechanisms in common, with the involvement of pro-inflammatory mediators. A bidirectional relationship between type 2 diabetes and periodontitis has been documented in several studies. Significantly less studies have focused on the association between periodontal disease and type 1 diabetes. The aim of the study is to analyze the association between periodontal status and type 1 diabetes mellitus. The “Preferred Reporting Items for Systematic Reviews and Meta-Analysis guidelines” was used and registered at PROSPERO. The search strategy included electronic databases from 2012 to 2021 and was performed by two independent reviewers. According to our results, we found one article about the risk of periodontal diseases in type 1 diabetes *mellitus* subjects; four about glycemic control; two about oral hygiene; and eight about pro-inflammatory cytokines. Most of the studies confirm the association between type 1 diabetes *mellitus* and periodontal diseases. The prevalence and severity of PD was higher in DM1 patients when compared to healthy subjects.

## 1. Introduction

Type 1 diabetes *mellitus* (DM1), also known as insulin-dependent diabetes or juvenile diabetes, has an idiopathic or autoimmune cause in which there is a destruction of the pancreatic β-cells [1,2,3,4]. It can be diagnosed at any age, but this type of diabetes often manifests itself in children, adolescents, and young adults [4]. According to the American Diabetes Association, type 1 diabetes represents about 5–10% of patients with diabetes [5,6].

Several clinical studies suggest that diabetes *mellitus* is a risk factor in the prevalence, progression, and severity of periodontal disease (PD). According to some authors, periodontal disease is considered the sixth most common complication of diabetes [6,7,8,9]. 

PD is a chronic inflammatory disease that causes the destruction of the tissues that support the tooth. This inflammatory process is caused by the presence of Gram-negative bacteria, which accumulate along the tooth margin, promoting a chronic and progressive local inflammatory response [5,10,11,12,13,14]. Gingivitis and periodontitis are the two forms of periodontal disease. Gingivitis is a superficial inflammation of the periodontium in which there is no attachment loss. When left untreated, it can reach the deep periodontium, evolving to periodontitis, which is an irreversible inflammation of the periodontium with tissue destruction and bone resorption [12,15]. The consequent loss of support structure can lead to loss of tooth parts and systemic inflammation [11,16]. 

Diabetes *mellitus* and PD correspond to inflammatory diseases that have pathogenic mechanisms in common, with the involvement of pro-inflammatory mediators [17]. According to some studies, the presence of elevated levels of pro-inflammatory mediators in the gingival tissues of diabetic patients, such as IL1-β (interleukin 1 beta), tumor necrosis factor (TNF-α), IL-6 (interleukin 6), matrix metalloproteinases (MMPs), prostaglandins (PGs), nuclear factor-kappa B receptor activator ligand/osteoprotegerin relationship (RANK-L/OPG), and oxidative stress, plays an important role in the initiation and progression of periodontal disease [7,8,9,17,18]. 

Type 1 diabetes *mellitus* is associated with elevated levels of systemic markers of inflammation. The elevated inflammatory state in diabetes contributes to both microvascular and macrovascular complications, and hyperglycemia can result in the activation of pathways that enhance inflammation, oxidative stress, and apoptosis [19].

The level of glycemic control is of key importance in determining increased risk of periodontal disease. The glycated hemoglobin (HbA1c) test is also widely used for the detection and control of diabetes mellitus. This test determines the amount of glucose that is irreversibly bound to the hemoglobin molecule of red blood cells and which will remain bound throughout its lifetime, around 30 to 90 days. The normal value for hemoglobin HbA1c is less than 6.5%; the higher the glucose level, the higher the percentage of glycated hemoglobin [19,20].

Although there are already plenty of studies on PD and type 2 diabetes *mellitus* (DM2), studies on the relationship of PD and DM1 remain scarce. The main objective of this systematic review is to analyze the association between periodontal status and type 1 diabetes *mellitus* and evaluate the effects of glycemic control in type 1 diabetes *mellitus* subjects with periodontal disease. 

## 2. Materials and Methods

This systematic review was conducted from June 2022 to September 2022, according to the “Preferred Reporting Items for Systematic Reviews and Meta-Analysis guidelines” (PRISMA) [21] using the databases MEDLINE via PubMed and Cochrane Library, Web of Science, and Scopus (from January of 2012 to November of 2022). The search was also conducted using the following journals: Journal of Clinical Periodontology, Journal of Periodontology, and Periodontology 2000 via Wiley Online Library (2012 to present). The research strategy used was: (type 1 diabetes mellitus [MeshTerms]) and (periodontal disease [MeshTerms]); (type 1 diabetes mellitus [MeshTerms]) and (periodontitis [MeshTerms]); (type 1 diabetes mellitus [MeshTerms]) and (chronic periodontitis [MeshTerms]); (gingivitis [MeshTerms]) and (type 1 diabetes mellitus [MeshTerms]).

Records were screened by the title, abstract, and full text by two independent investigators. Studies included in this review matched all the predefined criteria according to PICOS (“Population”, “Intervention”, “Comparison”, “Outcomes”, and “Study design”). A detailed search flowchart is presented in the Results section. 

The study protocol for this systematic review was registered on the International Prospective of Systematic Reviews (PROSPERO), under number CRD42022385448.

The eligibility criteria were organized, using the PICO method, as follows: − P (population): Type 1 Diabetic patients;− I (intervention/exposure): Periodontal disease;− C (comparison): Patients without periodontal disease;− O (outcome): to analyze the association between type 1 diabetes mellitus and periodontal disease.

The inclusion criteria corresponding to the PICO’s questions were articles in English, Portuguese, or Spanish, articles related to DM1, and cross-sectional studies, case-control studies, cohort studies, and randomized controlled clinical studies. On the other hand, the exclusion criteria were articles without an abstract available, literature reviews and meta-analyses, expert opinions, letters to editor, conference abstracts, animal studies, and studies investigating DM2 exclusively. We also excluded inflammatory diseases, chronic liver disease, or articles related to any treatment that may modify study parameters such as antibiotics, immunosuppressants, or antiepileptic drugs.

### 2.1. Extraction of Sample Data

The data were collected by drawing up a results table, and the information was collected taking into consideration the study design and aim, the eligibility criteria, the study population (with sample size and age group or average age), the duration in months or years of the study as well as the follow-up period, and the outcome measures and results.

### 2.2. Study Quality and Risk of Bias

To assess the methodological quality of a study and to determine the extent to which a study has addressed the possibility of bias in its design, conduct, or analysis, we used the Joanna Briggs Institute (JBI) guidance 2017 for each type of study (cross-sectional, case-control, cohort studies, or randomized controlled trials) [22]. For each type of study, a different questionnaire was conducted using the answers Yes (Y), No (*n*), Unclear (UN), Not/Applicable (NA). Two independent examiners (R.C./M.R.) were used to demonstrate intra- and inter-examiner reliability. 

## 3. Results

In total, 2975 studies were initially identified, and after removing duplicates and excluding articles by title and abstract, we investigated in a full-text analysis (Figure 1).

Finally, 15 cohort studies were included in our meta-analysis; the characteristics of all included studies are presented in the Table 5.

Figure 1 shows the detailed selection strategy of the articles.

### 3.1. Characterization of the Sample for the Quality of the Study

Quality assessments are shown in Table 1 for cross-sectional studies, Table 2 for case-control Studies, Table 3 for randomized controlled trials, and Table 4 for cohort studies. 

The degree of quality of the studies on the relational index used and the number of positive responses to the questions are mostly high, including nine articles [7,11,20,23,24,25,26,27,28], although we can also find five studies with moderate evidence [3,9,18,29,30] and one of low quality [8].

**Table 1 jcm-12-01147-t001:** Joanna Briggs Institute Critical Appraisal Checklist for Analytical Cross-Sectional Studies.

Joanna Briggs Institute Critical Appraisal Checklist for Analytical Cross-Sectional Studies.	1. Were the Criteria for Inclusion in the Sample Clearly Defined?	2. Were the Study Subjects and the Setting Described in Detail?	3. Was the Exposure Measured in a Valid and Reliable Way?	4. Were Objective, Standard Criteria Used for Measurement of the Condition?	5. Were Confounding Factors Identified?	6. Were Strategies to Deal with Confounding Factors Stated?	7. Were the Outcomes Measured in a Valid and Reliable Way?	8. Was Appropriate Statistical Analysis Used?
Antonoglou et al. [18], 2013	Y	Y	Y	Y	N	UN	Y	Y
Dakovic et al. [9], 2013	Y	Y	Y	Y	N	UN	Y	Y
Poplawska-Kita et al. [7], 2014	Y	Y	Y	Y	Y	UN	Y	Y
Jindal et al. [29], 2015	Y	Y	Y	Y	N	UN	Y	Y
Lappin et al. [23], 2015	Y	Y	Y	Y	Y	Y	Y	Y
Ismail et al. [24], 2017	N	Y	Y	Y	Y	Y	Y	Y
Roy et al. [25], 2019	Y	Y	Y	Y	Y	Y	Y	UN
Dicembrini et al. [11], 2021	Y	Y	Y	Y	Y	Y	Y	Y
Jensen et al. [26], 2021	Y	Y	Y	Y	Y	UN	Y	Y

**Table 2 jcm-12-01147-t002:** Joanna Briggs Institute Critical Appraisal Checklist for Case Control Studies.

Joanna Briggs Institute Critical Appraisal Checklist for Case Control Studies.	1. Were the Groups Comparable other than the Presence of Disease in Cases or the Absence of Disease in Controls?	2. Were Cases and Controls Matched Appropriately?	3. Were the Same Criteria Used for Identification of Cases and Controls?	4. Was Exposure Measured in a Standard, Valid, and Reliable Way?	5. Was Exposure Measured in the Same Way for Cases and Controls?	6. Were Confounding Factors Identified?	7. Were Strategies to Deal with Confounding Factors Stated?	8. Were Outcomes Assessed in a Standard, Valid, and Reliable Way for Cases and Controls?	9. Was the Exposure Period of Interest Long Enough to be Meaningful?	10. Was Appropriate Statistical Analysis Used?
Zizzi et al. [27], 2013	Y	Y	Y	Y	Y	Y	N	Y	Y	Y
Linhartova et al. [28], 2018	Y	Y	Y	Y	Y	Y	N	Y	Y	Y
Keles et al. [30], 2020	Y	Y	Y	Y	Y	UN	UN	Y	Y	Y
Sereti et al. [3], 2021	Y	Y	Y	Y	Y	UN	UN	Y	Y	Y

**Table 3 jcm-12-01147-t003:** Joanna Briggs Institute Critical Appraisal Checklist for Randomized Controlled Trials.

Joanna Briggs Institute Critical Appraisal Checklist for Randomized Controlled Trials.	1. Was True Randomization Used for Assignment of Participants to Treatment Groups?	2. Was Allocation to Treatment Groups Concealed?	3. Were Treatment Groups Similar at the Baseline?	4. Were Participants Blind to Treatment Assignment?	5. Were Those Delivering Treatment Blind to Treatment Assignment?	6. Were Outcomes Assessors Blind to Treatment Assignment?	7. Were Treatment Groups Treated Identically Other than the Intervention of Interest?	8. Was Follow up Complete and If Not, Were Differences between Groups in Terms of Their Follow up Adequately Described and Analyzed?	9. Were Participants Analyzed in the Groups to Which They Were Randomized?	10. Were Outcomes Measured in the Same Way for Treatment Groups?	11. Were Outcomes Measured in a Reliable Way?	12. Was Appropriate Statistical Analysis Used?	13. Was the Trial Design Appropriate, and any Deviations from the Standard RCT Design (Individual Randomization, Parallel Groups) Accounted for in the Conduct and Analysis of the Trial?
Ajita et al. [8], 2013	N	Y	NA	N	NA	N	NA	N	N	NA	N	NA	NA

**Table 4 jcm-12-01147-t004:** Joanna Briggs Institute Critical Appraisal Checklist for Cohort Studies.

Joanna Briggs Institute Critical Appraisal Checklist for Cohort Studies.	1. Were the Two Groups Similar and Recruited from the Same Population?	2. Were the Exposures Measured Similarly to Assign People to both Exposed and Unexposed Groups?	3. Was the Exposure Measured in a Valid and Reliable Way?	4. Were Confounding Factors Identified?	5. Were Strategies to Deal with Confounding Factors Stated?	6. Were the Groups/Participants Free of the Outcome at the Start of the Study (or at the Moment of Exposure)?	7. Were the Outcomes Measured in a Valid and Reliable Way?	8. Was the Follow up Time Reported and Sufficient to Be Long Enough for Outcomes to Occur?	9. Was Follow up Complete, and If Not, Were the Reasons to Loss to Follow up Described and Explored?	10. Were Strategies to Address Incomplete Follow up Utilized?	11. Was Appropriate Statistical Analysis Used?
Sun et al. [20], 2019	Y	Y	Y	Y	Y	Y	Y	Y	NA	NA	Y

### 3.2. Characteristics of the Included Studies

From each eligible study included in the present systematic review, we collected data about general characteristics, such as study design and aim, inclusion and exclusion criteria, as well as the study population (with sample size and age group or average age), the duration in months or years of the study, as well as the follow-up period and the outcome measures and results (Table 5). 

According to our results, we found one article about the risk of periodontal diseases in type 1 diabetes *mellitus* subjects; four about glycemic control; two about oral hygiene; and eight about pro-inflammatory cytokines.

## 4. Discussion

The aim of this systematic review is to analyze the association between type 1 diabetes mellitus and periodontal disease. 

There is emerging evidence of a two-way relationship between diabetes *mellitus* and periodontal diseases, with diabetes increasing the risk of periodontitis and periodontal inflammation negatively affecting glycemic control [1,6,7].

According to our results, there seems to be an association between PD and DM1, and the prevalence and severity of PD was higher in DM1 patients when compared to healthy controls [7,8,11,20]. Sun et al. [20] confirmed that DM1 patients exhibited an increased risk of PD (aHR = 1.45; *p* < 0.001) when compared to non-diabetic patients. In addition, the hazard of developing PD was markedly increased in DM1 patients with increased annual emergency room visits and hospitalizations for their diabetes (adjusted hazard ratio (aHR) of 13.0 and 13.2, respectively, *p* < 0.001). Concerning the two specific types of PD, DM1 patients had a 1.47-fold higher risk to develop gingivitis (95% CI = 1.36–1.59) and 1.66-fold higher risk to develop periodontitis (95% CI = 1.41–1.96), when compared to non-DM1 subjects. People aged 20–40 had a lower incidence of gingivitis and a higher incidence of periodontitis than those aged <20 in both case and control groups [20]. 

### 4.1. Glycemic Control

The evidence suggests that the level of glycemic control is of key importance in determining increased risk of periodontal disease [7,19,31]. For this reason, periodontal literature used categorical values for Glycated Hemoglobin (HbA1c) as seen in the new consensus report of the 2017 World Workshop on the Classification of Periodontal and Peri-Implant Diseases and Conditions for the staging and grading of periodontitis [32]. From our results, four articles were found that support this theory [7,8,26,29]. The goal of research led by Dicembrini et al. [11] was to investigate the prevalence of PD in DM1 patients and its association with glycemic control and glucose variability. A significant correlation was found between mean Clinical Attachment Loss (CAL) and Glucose Coefficient Variation (CV) (r = 0.31, *p* = 0.002), but not with Glycated Hemoglobin (HbA1c) (r = 0.038 *p* = 0.673). Furthermore, mean Periodontal Probing Depths (PPD) were associated with CV but not with HbA1c (r = 0.27 and 0.044; *p* = 0.007 and 0.619, respectively). A positive correlation between the CV and DM1 was seen after adjusting for the main confounders. 

In another study, conducted by Jensen et al. [26], the worsening of glycemic control was associated with increased severity of early markers of periodontal disease in children and adolescents with DM1. The HbA1c was positively correlated with plaque index (PI) (Rho = 0.34; *p* = 0.002), gingival index (GI) (Rho = 0.30; *p* = 0.009), bleeding on probing (BOP) (Rho = 0.44; *p* = 0.0001), and periodontal probing depths (PPD) > 3 mm (Rho = 0.21; *p* = 0.06). 

Furthermore, Jindal et al. [29] investigated the relationship between the severity of PD and glycemic control in DM1 patients in a hospital-based study, and the DM1 patients with poor metabolic control (PMC) exhibited increased inflammation (*p* < 0.005), more dental plaque, and clinical attachment loss when compared to those with fair and good glycemic control (GMC).

The study of Ajita et al. [8] showed that the bleeding index was significantly higher in DM1 patients, suggesting greater susceptibility for PD. When comparing the poor metabolic control patients with ones with good metabolic control, significant differences were recorded in PPD (*p* < 0.001), Bleeding Index (BI) (*p* < 0.001), and Clinical attachment Loss (CAL) (*p* = 0.001). CAL, BI, and PPD were greater in DM1 patients than in non-DM1 patients (4.337 ± 0.648 vs. 2.300 ± 0.557, respectively, *p* = 0.001; 2.708 ± 0.390 vs. 1.760 ± 0.434 respectively, *p* < 0.001; and 6.337 ± 0.650 vs. 5.181 ± 0.705, respectively, *p* < 0.001). The results showed a correlation between the bleeding index and disease severity in patients diagnosed with diabetes in a short period of time (4–7 years) (1.760 ± 0.434). On the other hand, longer durations of DM1 were associated with greater CAL. 

Another study by Poplawska-Kita et al. [7] studied the role of hyperglycemia in the development of periodontal disease. According to their study, periodontitis was found in 57.9% of DM1 patients, including 59.5% of these with poor metabolic control, which highlights the relationship between glycemic control and the increased risk of periodontal disease in DM1 subjects. 

### 4.2. Advanced Glycated-End Products

The presence of chronic hyperglycemia is related to the increased production of Advanced Glycated-End products (AGEs). AGEs are implicated in suppressed collagen production by gingival and periodontal ligament fibroblasts [3,27]. In addition, the binding of AGEs to a receptor increases the production of pro-inflammatory mediators, such as interleukin-1 β (IL-1β), tumor necrosis factor α (TNF)-α, and interleukin-6 (IL-6), involved in periodontal destruction [19,33]. 

The study of Zizzi et al. [27] attempted to evaluate the expression of AGEs in Diabetes-Mellitus-associated periodontitis. According to their findings, AGE-positive cells were not found either in fibroblasts or in gingival inflammatory cell infiltrates in subjects of the control group and in the group of systematically healthy individuals affected by chronic periodontitis. On the other hand, in the group of subjects with DM1 affected by chronic periodontitis, there was found a positive correlation between the duration of DM and the percentage of AGE-positive cells in epithelium (r: 0.610; *p*: 0.012), vessels (0.635; *p*: 0.008), and fibroblasts (r = 0.589; *p*: 0.016). A positive association was found between gingival expression of AGEs and the duration of DM1. 

Periodontal disease is an inflammatory process caused by Gram-negative anaerobic bacteria that are present in bacterial plaque along the tooth margin, causing a chronic and progressive response. For this reason, the presence of inadequate oral hygiene might contribute to the development of periodontal inflammation and further tissue destruction [34]. According to our research, there are two studies that support that evidence [24,25]. The study of Ismail et al. [24] showed that children with DM1 exhibited significantly greater plaque deposits (*p* = 0.01), a higher mean plaque index (*p* < 0.01), and a greater percentage of sites with bleeding on probing (*p* > 0.05) when compared to non-diabetics. Furthermore, the study by Sereti et al. [3] showed that the mean of GI, BOP, and the number of sites with PI and GI score > 1 was markedly higher in the DM1 group as compared to the controls. Moreover, the results by Roy et al. [25] showed that the mean presence of plaque, GI, and BOP and the mean sites with GI score ≥ 1 were appreciably higher in the DM1 group than in the control group, which suggests that these subjects will be more susceptible to developing periodontitis in the future. However, concerning the diagnosis of periodontal disease, no significant differences were observed. Gingivitis was present in 68% of the diabetics and 60% of nondiabetic subjects. Concerning the presence of periodontitis, fourteen patients of the control group had a diagnosis of periodontitis against fifteen of the diabetics group. In a multivariable logistic regression, periodontitis was related mainly to age and BOP. When comparing the periodontal parameters between controls and diabetics in younger (<40 years old) and older (>40 years old) subjects, the younger diabetic subjects showed significantly more plaque (*p* = 0.004) and inflammation (GI *p* < 0.001) compared with their matched controls. In the older group, gingival inflammation was markedly higher in diabetic subjects compared with controls (*p* = 0.003). According to the authors, this difference in the gingival health of young vs. old DM1 subjects to their matched controls may provide diagnostic advantages and prevention opportunities to exploit. 

### 4.3. Pro-Inflammatory Mediators

After the inflammatory stimulation, the pro-inflammatory cytokines such as IL-1β, IL-6, interleukin-8 (IL-8), and TNF-α and other pro-inflammatory mediators like prostaglandin E2 (PGE2) and Matrix Metalloproteinase (MMP) and the receptor activator of nuclear factor kB ligand (RANKL), as well as T cell regulatory cytokines (interleukin 18- IL-18) will increase, and periodontal destruction will occur [3,17,19,23,30,35]. According to our research, there are eight studies that support that evidence [3,9,11,18,23,27,28,30]. The study by Keles et al. [30] targeted parameters such as gingival crevicular fluid IL-18 and TNF-α levels in diabetic children with gingivitis. The clinical periodontal parameters, gingival crevicular fluid IL-18 and TNF-α levels, were similar between diabetic and systemically healthy children (*p* > 0.05). The gingivitis subgroups showed a significantly higher PI, GI, PPD, GCF volume, and TNF-α total amounts than the healthy subgroups (*p* < 0.0001). However, the IL-18 concentrations were significantly higher in the periodontally healthy subgroups than in gingivitis subgroups. The TNF-α were positively correlated with PI, GI, PPD, GCF volumes, and IL-18 concentration (r = 0.552, *p* = 0.01; r = 0.579, *p* = 0.01; r = 0.534, *p* = 0.01, respectively). However, there was a negative correlation between the IL-18 concentration and the TNF-α (−0.524, *p* = 0.01). It is known that the presence of TNF-α in periodontal tissues acts as a risk factor for the beginning of alveolar bone destruction and periodontal connective tissue breakdown by increasing both secretion of matrix metalloproteinases and osteoclast formation [30,36]. The increased gingival crevicular fluid (GCF) TNF-α in DM1 children with gingivitis confirms that TNF-α is closely related to gingival inflammation [30]. 

The IL-18 belongs to the IL-1 superfamily and has been implicated in the pathogenesis of chronic diseases, including DM1. According to the authors, despite of the fact that previous studies have reported that serum IL-18 levels in diabetic children were higher when compared to healthy controls, there is no evidence of the GCF IL-18 levels from diabetic and non-diabetic children [30]. 

In another study by Poplawska-Kita et al. [7], the GMC group showed the lowest concentration of C-Reative Protein (CRP) and TNF-α among all groups. DM1 patients with periodontitis showed higher fibrinogen (371.3 ± 114.7, *p* < 0.01) and TNF-α (1.6 ± 1.2, *p* < 0.001) concentrations, as well as lower OHI (2.1 ± 0.7, *p* < 0.001) and a lower number of teeth (*p* < 0.001). The number of sextants without signs of periodontal disease (CPI 0) was correlated negatively with fibrinogen (r = −0.272; *p* < 0.05) and TNF-α (r = −0.233; *p* < 0.05) levels. The evidence suggests that the CRP and fibrinogen are produced in response to the action of pro-inflammatory cytokines and are responsible for a systemic response [26]. The number of sextants with 4–5 mm deep pathologic pockets (CPI 3) was correlated positively with TNF-α (r = 0.348; *p* < 0.01) and fasting glucose level (r = 0.217; *p* < 0.05). Taken together, their results suggest a role for TNF-α in periodontal destruction, especially in those with poor metabolic control, and inadequate oral hygiene might contribute to the development of inflammation and further tissue destruction. 

The study of Linhartov et al. [28] targeted parameters like IL-8 plasma levels in patients with DM1 and systematic health controls. According to their findings, concentrations of circulating IL-8 levels were not significantly associated with the level of glycemic control (blood glucose and HbA1c), smoking status, and clinical parameters like GI, PPD, and attachment loss (AL) (*p* > 0.05). However, patients with DM1 showed higher circulating IL-8 plasma levels than Health Control with Chronic Periodontitis/non-periodontitis Heath Control. The IL-8 is involved in the initiation and amplification of a severe inflammatory reaction, and it is secreted by several cell types in response to inflammatory stimuli [3,9,23,28]. Furthermore, there were statistically significant differences between the non-periodontitis healthy control in comparison to the group with Chronic Periodontitis and DM1 with chronic periodontitis concerning the GI: (0.3 ± 0.2) vs. (0.9 ± 0.3) and (1 ± 0.3), respectively (*p* < 0.01), and numbers of sites and teeth with a pocket depth ≥ 5 mm and attachment loss ≥ 5 mm (*p* < 0.01), which means that DM1 with chronic periodontitis showed a greater inflammation and clinical attachment loss that is related to periodontal destruction. 

Sereti et al. [3] evaluated the GCF levels of MMP-8, IL-8, and AGEs in DM1 patients with different glycemic levels and compared them with healthy controls. The median GCF levels of MMP-8 (control: 38.3 µg/mL vs. DM1 group: 32.1 3 µg/mL, *p* = 0.538), IL-8 (control: 225 pg/mL vs. DM1 group: 220 pg/mL, *p* = 0.433), and AGEs (control: 5.8 µg/mL vs. DM1 group: 3.4 µg/mL, *p* = 0.905) did not differ significantly. Concerns the presence of GCF markers, no significant differences were observed between younger diabetics (<40 years old) and controls or between older diabetics (>40 years old) and controls, even when the groups were divided according to glycemic control. According to the evidence, the MMP-8 is associated with pathologic extracellular matrix destruction and is the main collagenase that is found in inflamed gingiva in adult periodontitis [3]. 

In another study, Lappin et al. [23] compared the circulating levels of IL-6 and IL-8 in patients with DM1 with and without periodontitis. The evidence suggests that circulating levels of IL-6 are implicated in poor clinical outcomes in DM1 and susceptibility to periodontal disease [23]. However, no difference was seen in IL-6 plasma levels between groups. On the other hand, the plasma levels of IL-8 were higher in the periodontitis group when compared to the healthy group (*p* < 0.001). The DM1 group and the DM1 group with periodontitis exhibited higher levels of IL-8 than healthy volunteers (*p* < 0.001, for both). Patients with DM1 with Periodontitis showed higher levels of IL-8 when compared to patients with periodontitis (*p* < 0.05).

Furthermore, Dakovic et al. [9] investigated the differences between the salivary levels of IL-8 in patients with DM1 with or without concomitant periodontitis and healthy patients. According to their findings, DM1 patients exhibited a significantly higher level of salivary IL-8 when compared to the control group (*p* < 0.005). However, there were no differences in the level of salivary IL-8 between DM1 patients with periodontitis and DM1 patients without periodontitis. There was a statistically significant difference for PPD, CAL, and BOP between DM1 patients with periodontitis and DM1 without periodontitis (*p* < 0.05). The correlation between IL-8 and clinical parameters in DM1 children did not show any statistically significant correlation. 

Another aspect worth considering is the circulating levels of RANKL and osteoprotegerin (OPG) in the extent of periodontal destruction. According to the literature, the OPG and RANKL have been suggested to play an important role in the differentiation of osteoclasts and, furthermore, in periodontal-disease-associated bone loss [18]. The study by Antonoglou et al. [18] showed that DM1 patients with no or mild periodontitis had a total of 16 sites (16.4 ± 14.5) presented with bleeding and PPD ≥ 4 mm and 0.7 sites (0.7 ± 1.0) with attachment loss (AL) ≥ 4 mm. When compared to severe periodontitis, the corresponding figures were (39.6 ± 21.9) and (38.8 ± 18.5) respectively, which suggest that PPD and AL increase with the severity of periodontal disease in DM1 subjects. 

The OPG was 135 pg/mL in subjects with severe periodontitis and 96.0 pg/mL in those with no or mild periodontitis. The results showed a positive association between AL ≥ 4 mm and severity of periodontitis and the level of serum OPG. However, when the analyses included only non-smokers, the positive association mentioned above showed a major drop in the strength and statistical significance. The results did not find any association between serum RANKL level or RANKL/OPG ratio and periodontal variables. The RANKL in the group of subjects with no or mild periodontitis was 18.1 pg/mL and 33.2 pg/mL for those with severe periodontitis. Concerning the RANKL/OPG ratio, the values were (0.2 ± 0.1) for the first group (no or mild periodontitis) and (0.1 ± 0.1) for those with severe periodontitis. According to their study, the serum OPG, which is a marker of systemic inflammatory burden, could also be an indicator of periodontal tissue destruction in DM1 subjects [18]. 

When evaluating the quality of the studies, using the Joanna Briggs method, most of them were classified as high or moderate score quality. The item that presented the worst results was the “strategies to deal with confounding factors state”, with many of them not presenting such cofactors such as tobacco habits. Unfortunately, there are no other systematic review to our knowledge to make a comparison to our results regarding the quality of the studies (or existence of bias).

Our systematic review found that the interplay between the two conditions highlights the importance of the need for a good communication between the endocrinologist and dentist about diabetic patients, always considering the probability that the two diseases may occur simultaneously in order to ensure the early diagnosis of both. 

We acknowledge some limitations in our systematic review, firstly related with the few existing original articles suitable for inclusion (such as randomized controlled trials), or lack of information that could been used for a quantitative analysis. For these reasons, a meta-analysis was not possible. Nevertheless, this study, to our knowledge, is an original systematic review of the existing data regarding the association between DM1 and PD. 

## 5. Conclusions

Most of the studies confirm the association between DM1 and PD. The prevalence and severity of PD was higher in DM1 patients when compared to healthy subjects. 

Periodontal disease was associated with glucose variability in DM1 patients. Furthermore, an increased periodontal inflammatory tendency corresponded to those individuals with poor metabolic control. DM1 subjects with increased HbA1c levels were associated with an increase in plaque index, gingival index, probing depths > 3 mm, and clinical attachment loss when compared to healthy subjects. According to some studies, longer durations of DM were associated with greater periodontal attachment loss.

In addition, subjects with DM1 showed higher levels of IL-8, TNF-α, CPR, and fibrinogen. However, the findings of this review showed that research having a prospective longitudinal design should be done to clarify the roles of the pro-inflammatory cytokines in periodontal disease.

## Figures and Tables

**Figure 1 jcm-12-01147-f001:**
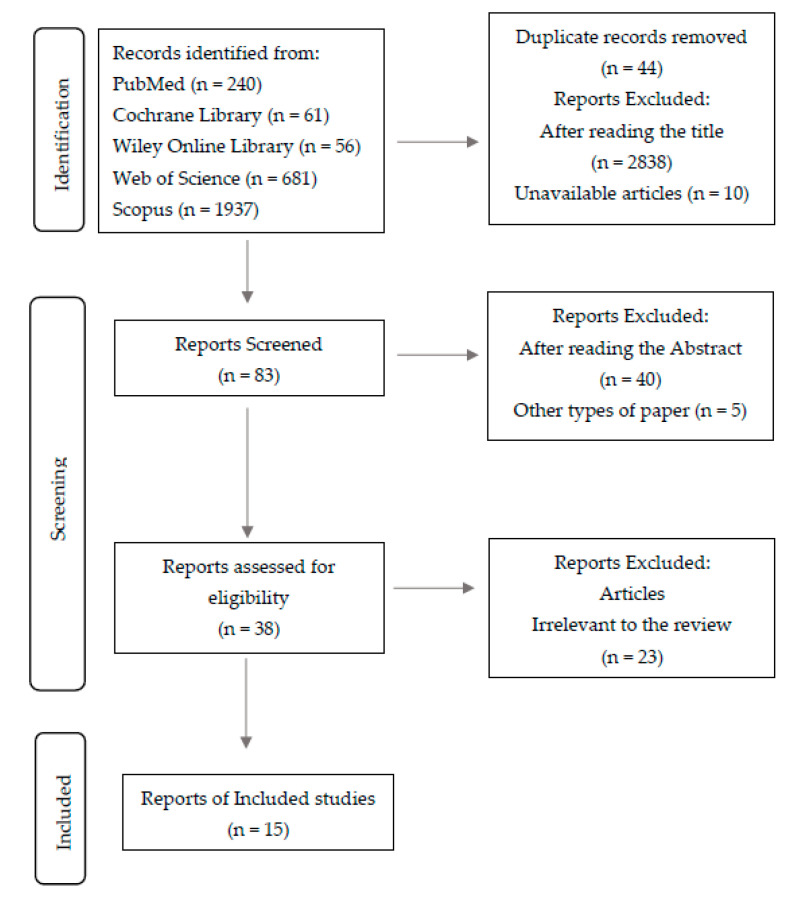
Flow diagram of study selection.

**Table 5 jcm-12-01147-t005:** The main characteristics of the included studies.

Authors	Study Design	Study Aim	Inclusion Criteria	Exclusion Criteria	Sample Size	Age Group	Study Duration	Outcome Measures	Results
Antonoglou et al. [18], 2013	Cross-sectional study	To explore the associations between the extent of periodontal destruction and circulating levels of RANKL and OPG.	Subjects from a primary health care diabetes unit in the City of Oulu, Finland and others from the Clinic of Internal Medicine, Oulu University Hospital, Oulu, Finland. - Subjects examined clinically by a periodontal specialist at the Specialist Dental Health Care Unit, Oulu (Finland).	Subjects needing prophylactic antibiotic medication in association with periodontal probing; subjects who used immunosuppressive medication or had had antibiotics during the past 4 months.	80 DM1 patients (46 Female; 34 Male) No or mild periodontitis (*n* = 40) Moderate periodontitis (*n* = 28) Severe periodontitis (*n* = 12)	18–74 years (38 ± 12.3)	NR	- AL (Attachment Loss) - OPG - RANKL - Duration of DM	Subject characteristics presented as mean values/subject (±SD) in different periodontal disease categories: Patients with no or mild periodontitis: - Total of 16 sites (16.4 ± 14.5) presented with bleeding and PPD ≥ 4 mm - 0.7 sites (0.7 ± 1.0) with AL ≥ 4 mm. - OPG: 96.0 pg/mL - The sRANKL: 18.1 pg/mL - The sRANKL/OPG ratio: 0.2 ± 0.1 Patients with severe periodontitis: - Total of 16 sites (39.6 ± 21.9) - sites with attachment loss (AL) ≥ 4 mm (38.8 ± 18.5) - OPG was 135 pg/mL - The RANKL—33.2 pg/mL - The RANKL/OPG ratio: 0.1 ± 0.1 The results showed a positive association between AL ≥ 4 mm and severity of periodontitis and the level of serum OPG. - The results did not find any association between serum sRANKL level or sRANKL/OPG ratio and periodontal variables. - This study showed a strong correlation between the age of the subjects and the duration of DM (r = 0.530, *p* ≤ 0.001).
Ajita et al. [8], 2013	Randomized, controlled clinical study	To determine the relationship between DM1 an PD and to analyse how diabetes metabolic control, complications and duration are related with periodontal parameters.	Subjects aged between 18 and 50 years; DM1 patients diagnosed for more than 3 years; Subjects without any active infection; Individuals with >14 natural teeth present and at least 5 teeth with PPD ≥ 5 mm and CAL ≥ 3 mm and who had not had any periodontal treatment in the last 6 months.	If they had non-type 1 DM; Pregnancy and lactation; Subjects with any inflammatory disease, chronic liver disease, or patients taking antibiotics, immunosuppressants, and antiepileptics.	DM1 patients (*n* = 20 [14 Males/6 Females]) Non-DM1 (*n* = 20 [14 Males/6 Females])	18–50 years	NR	- PPD - CAL - BI - Duration of DM	- BI: significantly higher in DM1 patients (2.708 ± 0.390) (mean ± SD) when compared to non-diabetic (1.760 ± 0.434). Relationship between periodontal parameters in PMC patients vs. GMC patients: (mean ± SD) PPD (6.429 ± 0.723) vs. (5.814 ± 0.693) BI (2.646 ± 0.402) vs. (1.129 ± 0.362) CAL (4.356 ± 0.688) vs. (2.214 ± 0.679) significant differences were recorded in PPD (*p* < 0.001), BI (*p* < 0.001), and CAL (*p* = 0.001) between those groups. - DM1 patients vs. non-diabetic patients: (mean ± SD) CAL (4.337 ± 0.648) vs. (2.300 ± 0.557) (*p* = 0.001) BI (2.708 ± 0.390) vs. (1.760 ± 0.434) (*p* < 0.001) PPD (6.337 ± 0.650) vs. (5.181 ± 0.705) (*p* < 0.001) - The results showed a correlation between the bleeding index and disease severity in patients diagnosed with diabetes in a short period of time (4–7 years) (1.760 ± 0.434).
Dakovic et al. [9], 2013	Cross-sectional study	To investigate the differences between the salivary levels of IL-8 in patients with DM1 with or without concomitant periodontitis and healthy patients.	- Patients attending the Outpatient Diabetes Clinic at the Mother and Child Healthcare Institute of Serbia, over 5 month period. - DM1 patients treated only with multiple daily insulin injections. - Healthy patients aged 7–18 attending the Clinic of Dental Medicine, Military Medical Academy (Belgrade) for a dental check-up.	Children that were undergoing active orthodontic therapy, had other systemic disease, or had received systemic antibiotic therapy in 6 months prior to the study.	Children DM1 (*n* = 20 [9 Males/11 Females]): - with periodontitis (*n* = 10) - without periodontitis (*n* = 10) Healthy children and adolescents *N* = 20 [8 Males/12 Females])	7–18 years	NR	- PPD - BOP - CAL - Salivary IL-8 level	Periodontal measurements in DM1 children vs. healthy children (Group control): (mean ± SD) CAL (0.89 ± 0.57) vs. (0.89 ± 0.24) (*p* = 0.95) PPD (1.69 ± 0.41) vs. (1.45 ± 0.32) (*p* = 0.05) BOP (0.65 ± 0.33) vs. (0.26 ± 0.28) (*p* = 0.0001) - PPD and BOP were substantially higher in DM1 group compared to the healthy patients group. - Periodontal measurements in DM1 children with periodontitis vs. DM1 children without periodontitis (mean ± SD) - CAL (1.31 ± 0.49) vs. (0.47 ± 0.22) (*p* = 0.0001); - PPD (2.05 ± 0.18) vs. (1.33 ± 0.19) (*p* = 0.0001); - BOP (0.88 ± 0.33) vs. (0.43 ± 0.56) (*p* = 0.0004) There was a statistically significant difference for PPD, CAL, and BOP between DM1 subjects with periodontitis and DM1 without periodontitis. - DM1 patients exhibited a significantly higher level of salivary IL-8 when compared to the control group (*p* < 0.005). No differences in the level of salivary IL-8 between DM1 patients with periodontitis and DM1 patients without periodontitis. Correlations between clinical parameters and salivary IL-8 levels in DM1 children PPD (r = 0.07, *p* = 0.78); CAL (r = 0.04, *p* = 0.85), BOP (r = −0.19, *p* = 0.43) A correlation between the levels of salivary IL-8 and clinical parameters was not found.
Zizzi et al. [27], 2013	Case-Control Study	To evaluate the expression of AGEs in DM-associated periodontitis.	Age > 35 years; the presence of at least 20 teeth; - For periodontitis subjects a diagnosis of generalized, severe, chronic periodontitis made on the basis of the presence of more than 30% of measured sites with >5 mm of CAL; - For DM patients, the diagnosis of the type of DM is made at least 12 months before the study; - For nondiabetic subjects, HbA1c in the nondiabetic range (<6.1%) and plasma glycemia lower than 100 mg/dL; - For healthy subjects, PD < 3 mm, GI = 0 without clinical inflammation and CAL < 2 mm.	Presence of any important disease other than DM in the groups of diabetic subjects; Being smoker; having taken antibiotics, corticosteroids, or nonsteroidal anti-inflammatory drugs within the 6 months before treatment; having undergone periodontal treatment within the previous 2 years.	Healthy subjects (CT, *n* = 16 [12 Male/4 female]); Subjects DM1 suffering from generalized, severe CP (PD-DM1, *n* = 16 [11 Male/5 Female]) Systematically healthy individuals affected by periodontitis (PD-S, *n* = 16 [9 Males/7 females].	CT: 55 ± 1.76 * *p* < 0.05 PD-S 56.5 ± 1.32 * *p* < 0.05 PD-DM1 46.1 ± 0.70 * *p* < 0.05 vs. PD-DM1	2005–2011	- PPD - CAL - BL - AGEs	Periodontal parameters of CT vs. PD-S vs. PD-DM1 subjects: (median [interquartile range, IQR] GI: ((0) [0–0]) vs. (1.6 [1.4–2.3]) and (1.9 [1.4–2.6]) PPD (2.6 [2.2–2.8]) vs. (7.1 [7–7.3]) and (6.9 [6.9–7.1], CAL (1.1 [0.8–1.3]) vs. (6.6 [6.4–6.7]) and (6.6 [6.2–6.7] BL (5 [4–6.7]), vs. (60.5 [58.2–62]) and (59 [58.2–62.7]) There were statistically significant differences between healthy subjects (CT) in comparison to the group of systematically healthy individuals affected by periodontitis (PD-S) and DM1 subjects affected by periodontitis (PD-DM1) (*p* < 0.05) - PD-DM1: epithelium AGE % (90 [75–93.7]); [IQR] vessels AGE % (74 ± 2.38) (mean ± SD) - CT: Epithelium AGE% (62.5 [46.2–73.7]) (*p* < 0.05) Vessels AGE % (51.8 ± 2.88) (*p* < 0.05) - PD-S: Epithelium AGE% (70 [61.2–70]) (*p* < 0.05) Vessels AGE% (58.7 ± 4.19) (*p* < 0.05) On the gingival tissue from PD-DM1, there was found a significant increase in the number of AGE-positive cells in the epithelium and in vessels when compared to the CT and PD-S group. - AGE-positivity cells were not fund in fibroblast and in inflammatory infiltrates in subjects of the CT and PD-S group. - A positive correlation was found in PD-DM1 subjects between the duration of DM and the percentage of AGE-positive cells in epithelium (r: 0.610; *p*: 0.012), vessels (0.635; *p*: 0.008), and fibroblasts (r = 0.589; *p*: 0.016).
Popławska-Kita et al. [7], 2014	Cross-sectional study	The role of hyperglycemia in the development of periodontal disease.	NR	Presence of systemic diseases other than DM1; subjects taking immunosuppressive drugs, steroids, or non-steroidal anti-inflammatory drugs, pregnancy and fixed orthodontic appliances.	- 40 Subjects Group Control (GC) According to the metabolic control: - 107 DM1 subjects DM1+HbA1c ≤ 6.5% (*n* = 22 [4 Males/14 Females]) DM1+HbA1c ≥ 6.5%; (*n* = 85 [50 Males/35 Females]) According to the presence of periodontitis: DM1+No-periodonti tis (*n* = 45) DM1+Periodontitis (*n* = 62) GC+No-periodontitis (*n* = 3 4) GC+periodontitis (*n* = 6)	According to the metabolic control: DM1 subjects with HbA1c ≤ 6.5%: 34.8 ± 10.9 HbA1c ≥ 6.5%; 37.9 ± 3.7 Control: 32.3 ± 1.0 According to the presence of periodontitis: DM1+periodontitis: 42 ± 12.7 DM1+no-periodontitis: 30.7 ± 11.1 GC+no-periodontitis: 29.4 ± 9.5 GC+periodontitis: 48 ± 2.1	NR	- HbA1c - CRP - (TNF)-α - Fibrinogen - OHI	Periodontitis was found in: - 15% of the controls - 57.9% of DM1 patients - 59.5% of DM1 with PMC. The incidence of periodontitis is increased in DM1, especially in those with poor metabolic control Biochemical characteristics of the GMC vs. PMC vs. CT (mean ± SD) CRP (ng/mL) (4.8 ± 1.2) vs. (10.9 ± 23.2) vs. (7.1 ± 8.5) TNF-α (pg/mL) (1.0 ± 0.6) vs. (1.25 ± 1.06) vs. (1.5 ± 1.6) GMC had the lowest concentration of CPR among all groups. The clinical characteristics of GMC vs. PMC (mean ± SD) - HbA1c (6.0 ± 0.6) (*p* < 0.01) vs. (9.8 ± 2.4) - Fasting glucose level (mg/dl) (126 ± 60.9) (*p* < 0.05), vs. (172.83 ± 72.4) (*p* < 0.01); PMC group exhibited significantly higher HbA1c and fasting glucose level. DM1 with periodontitis showed: Higher: (mean ± SD) - Fibrinogen (371.3 ± 114.7) (*p* < 0.01) - TNF-α (1.6 ± 1.2) (*p* < 0.001) - OHI (2.1 ± 0.7) Lower: - Teeth number (*p* < 0.001) - CPI 0/fibrinogen (r = −0.272; *p* < 0.05) - CPI 0/TNF-α (r = - 0.233; *p* < 0.05) - CPI 3/TNF-α (r = 0.348; *p* < 0.01) - CPI 3/fasting/glucose (r = 0.217; *p* < 0.05) The number of sextants without signs of periodontal disease (CPI 0) was correlated negatively with fibrinogen (*p* < 0.05), whereas the number of sextants with 4–5 mm deep pathologic pockets (CPI3) were correlated positively with TNF-α (*p* < 0.01) and fasting glucose level (r = 0.217; *p* < 0.05)
Jindal et al. [29], 2015	Cross-sectional study	To investigate the relationship between severity of periodontal disease and glycemic control in patients with DM1 in a hospital-based study.	Age between 12 and 25 years and with diagnosis of DM1 for more than 3 months duration.	Patients non-DM1, undergoing active orthodontic treatment; patients with any chronic inflammatory disease and on long-term medications that could influence the studied parameters such us antibiotics and antiepileptic or immunosuppressive drugs.	50 DM1 patients [32 Males/18 females]: - Group A- Good (HbA1c ≤ 7) *n* = 15 - Group B- Fair (HbA1c = 7–8) *n* = 16 - Group c- Poor (HbA1c > 8): 19	Between 12 and 25	NR	- PPD - CAL - PI - GI	Mean standard derivation of periodontal parameters between: GMC vs. Fair Metabolic control vs. PMC: (mean ± SD) - PPD (2.93 ± 0.59) vs. (3.81 ± 0.75) vs. (5.31 ± 0.20) - CAL (3.33 ± 0.48) vs. (4.43 ± 0.62) vs. (6.15 ± 1.38) - PI (1.25 ± 0.20) vs. (1.82 ± 0.45) vs. (2.39 ± 0.18) - GI (1.25 ± 0.34) vs. (1.43 ± 0.33) vs. (2.01 ± 0.29) (*p* < 0.05) DM1 with poor metabolic control exhibited increased GI, PI, PPD, and CAL when compared to other groups.
Lappin et al. [23], 2015	Cross-sectional study	To compare circulating levels of IL-6, IL-8 and CXCL5 in patients DM1, with or without periodontitis to control groups of systemically healthy, non-smoking, individuals with and without periodontitis. To determine the effect of AGE, in the presence and absence of *Pg* LPS, on IL-6, IL-8 and CXCL5 expression by THP-1 monocytes and OKF6/TERT-2 cells.	Diabetic subjects were diagnosed by trained clinicians and had been attending the outpatient clinic for monitoring of glycated hemoglobin for more than 2 years. In the periodontitis group, the participants had to have a minimum of two sites with probing depth and attachment loss ≥ 5 mm. None of the subjects were receiving periodontal treatment at the time of diagnosis.	No history of smoking within the past 5 years; pregnancy at the time of the recruitment; taking immunosuppressive drugs antibiotics or anti-inflammatory drugs within 6 weeks of recruitment; individual with less than 20 teeth and subjects who were unable to consent.	104 Subjects: Healthy volunteers (H *n* = 19 [63%Males/37%females]. Patients with periodontitis (PD *n* = 23 [46%Males/54% females]. DM1 patients (DM1n = 28 [36% Males/64% females]. DM1 patients with periodontitis (DM1+*p n* = 34 [45%Males/55%females].	H: 33 ± 8 PD: 40 ± 11 DM1: 35 ± 10 DM1+P: 36 ± 9	NR	- PPD - AL - BOP - Plasma IL-8 levels - Plasma IL-6 levels	- HbA1c: (mean ± SD) DM1+P (73.8 ± 17.0) DM1 (71.6 ± 16.3) H (32.2 ± 1.1) PD (33.3 ± 1.1) - Diabetic patients with or without periodontitis showed higher levels of glycated hemoglobin when compared to the healthy group or even with the group of periodontitis subjects. - Not significant difference between the diabetic and non-diabetic groups: The mean of sites with PPD of ≥5 mm, the number of teeth with PPD ≥ 5 mm, number of sites with AL ≥ 5 mm, teeth with AL ≥ 5 mm and the proportion of sites with BOP. - Plasma IL-6 levels did not o differ between the four groups; - Plasma levels of IL-8 were higher in periodontitis group, when compared to the healthy group (*p* < 0.001). - DM1 group and DM1+*p* group exhibited higher levels of IL-8 than healthy volunteers (*p* < 0.001, for both); - Patients with DM1+*p* showed higher levels of IL-8 when compared to patients with periodontitis (*p* < 0.05).
Ismail et al. [24], 2017	Cross-sectional study	To compare the caries experience and periodontal health status between children with DM1 and healthy age- and sex-matched controls.	DM1 patients, who are members of the Honk Kong Juvenile Diabetes Association; People that sign the consent form;	Patients who did not have any systemic disease or problems with manual dexterity; were not undergoing active orthodontic treatment, and had not received any dental treatment for the past 1 year.	64 Children: - DM1 (*n* = 32 [16 Males/ 16 Females]) - Control group (*n* = 32 [16 Males/ 16 Females])	(12 ± 4 years)	NR	- Plaque - GI - Gingivitis - PI - BI - CI - HbA1c	Periodontal health status between DM1 vs. Non-DM1: Mean (SD) Plaque: 0.66 (0.46) vs. 0.43 (0.16) (*p* = 0.01) PI: 0.76 (0.40) vs. 0.46 (0.14) (*p* < 0.01) BI: 0.20 (0.18) vs. 0.16(0.11) (*p* > 0.05) CI: 0.14(0.15) vs. 0.13(0.15) (*p* > 0.05) Gingivitis: 0.50(0.35) vs. 0.51(0.22) (*p* > 0.05) The Children with DM1 exhibited significantly greater plaque deposits (*p* = 0.01), a higher mean plaque index (*p* < 0.01), also had a greater percentage of sites with bleeding on probing (*p* > 0.05), when compared to non-diabetics. The percentage of sites with calculus deposits and gingivitis was similar in both groups (*p* > 0.05)
Linhartova et al. [28], 2018	Case-Control Study	To determine IL-8 plasma levels; IL-8 (−251A/T, rs4073) and its receptor 2 (CXCR2, +1208C/T, rs112679) polymorphisms; the presence of the selected bacteria in DM1 and DM2 patients and systemically healthy controls (HC) with periodontal status.	- The willingness to participate, compliance with the diagnostic criteria for Chronic periodontitis and or Diabetes *Mellitus*, and for the control group systemic and periodontal health. - Patients examined by a periodontist and did not receive any treatment before measuring periodontal indices.	- Patients that declined the periodontal treatment for periodontitis; - patients having immunodeficiency disorders, current pregnancy or lactation, immunosuppression attributable to medication or current illness; taking antibiotics or anti-inflammatory drugs with 6 weeks of recruitment; - Subjects with <20 teeth (only in healthy controls) and the inability to consent.	153 Patients: - DM1 subjects+CP (n = 36, 44.4% Males) -DM2 patients with CP (40.9% Males) - From HC+CP (n = 32, 26.8% Males) - Non-periodontitis NP-HC (*n* = 41, 31.3% Males)	The mean age was similar for patients with DM1+CP and HC	NR	- GI - PPD - AL - IL-8 plasma levels	Clinical periodontal parameters between NP-HC vs. HC+CP vs. DM1+CP: (mean ± SD) GI (0.3 ± 0.2) vs. (0.9 ± 0.3) vs. (1 ± 0.3) (*p* < 0.01) N of sites with PPD ≥ 5 mm (0) vs. (18 ± 17) vs. (20 ± 19) (*p* < 0.01) N of teeth with PPD ≥ 5 mm (0) vs. (10 ± 7) vs. (11 ± 7) (*p* < 0.01) N of sites with AL ≥ 5 mm (0) vs. (32 ± 21) vs. (38 ± 28) (*p* < 0.01) N of teeth with AL ≥ 5 mm (0) vs. (15 ± 7) vs. (15 ± 7) (*p* < 0.01) - HC+CP vs. DM1+CP, similar numbers were found (*p* > 0.05) There were statistically significant differences between non-periodontitis HC in comparison to HC+CP and DM1+CP in which concerns the GI (*p* < 0.01); and numbers of sites and teeth with a pocket depth ≥5 mm and attachment loss ≥ 5 mm (*p* < 0.01). - IL8 plasma levels and clinical parameters: median [interquartile range, IQR] DM1+CP vs. HC+CP: 15.09 pg/mL [9.73–20.32] vs. 11.02 pg/mL [6.47–15.17], (*p* ≤ 0.05) - NP- HC vs. HC+CP, (10.53 pg/mL [8.48–12.58] vs. 11.02 pg/mL [6.47–15.17] DM1 patients had significantly higher levels of IL-8 than did HC+CP individuals (*p* ≤ 0.05). The groups of non-periodontitis HC and HC+CP, exhibited similar IL-8 plasma levels. - IL-8 plasma levels in DM1 GMC vs. DM1 PMC: 12.68 pg/mL [10.52–40.56] vs. 14.04 pg/mL [10.05–19.67], (*p* > 0.05) Concentrations of circulating IL-8 levels were not significantly associated with the level of glycemic control (blood glucose and HbA1c and clinical parameters like GI, PPD and AL (*p* > 0.05). However, patients with DM1 showed higher circulating IL-8 plasma levels than HC+CP/non-periodontitis HC.
Roy et al. [25], 2019	Cross-sectional study	To evaluate the periodontal clinical conditions and oral health behaviour in a cohort of subjects DM1 and in a control group matched for age, sex and smoking status.	Subjects diagnosed with DM1 for more than a year and have at least 10 natural remaining teeth.	Individuals who had taken antibiotics in the previous 3 months. Subjects with history of systemic disease like cancer, HIV, bone metabolic disease, history of radiation or immunosuppressive/modulating therapy; disorders that compromise wound healing.	Patients group control: (*n* = 50 [30 Males/ 20 Females]) Patients DM1: (*n* = 50 [30 Males/ 20 Females])	18–85 year-aged	July 2016–July 2018	- GI - PI - PPD - BOP - CAL - REC - HbA1c	Dental examination results of Control vs. DM1: (%±SD) GI: 0.4 (0.4) vs. 1.1 (0.7) (*p* = 0.000) BOP: 29.4 (16.4) vs. 40.5 (22.2) (*p* = 0.009) Number of sites PI > 1, 13.8 (14.5) vs. 23.9 (27.2) (*p* = 0.047) Number of sites GI > 1, 18.8 (23.1) vs. 59.2 (57.6) (*p* = 0.001) The mean presence of plaque, GI, BOP, and the mean sites with GI score ≥ 1, were appreciably higher in DM1 group than in the control group. - PPD, REC, AL and the mean number of sites with a PI score of ≥1 and mean number of sites with PPD > 4 mm that bleed upon probing did not differ between the groups. Periodontal status Control vs. DM1: n (%) Gingivitis: 30 (60.0) vs. 34 (68.0) Periodontitis: 14 (28.0) vs. 15 (30.0) Gingivitis was present in 68% of the diabetics. - Periodontal parameters between controls vs. diabetics: mean ± SD younger (<40 years old) subjects PI 0.3 (0.2) vs. 0.6 (0.4) (*p* = 0.004) GI 0.3 (0.3) vs. 1.1 (0.7) (*p* = 0.000) Older (>40 years old) subjects PI 0.4(0.3) vs. 0.5(0.4) (*p* = 0.260) GI 0.5(0.4) vs. 1.0(0.6) (*p* = 0.003) Diabetics <40 years old had significantly more plaque (*p* = 0.004) and inflammation (GI; *p* < 0.001) compared with their matched controls. In the older group (>40 years old), gingival inflammation was markedly higher in diabetic patients compared with controls (*p* = 0.003) Mixed effects logistic regression for odds of periodontitis: Univariable OR [95% CI] Age 1.10 [1.05, 11.17], (*p* < 0.001) BOP 1.04 [1.02, 1.09], (*p* = 0.009) The only variables identified as determinants of the periodontal condition in both diabetic and control were age (*p* < 0.001) and BOP (0.009). Mixed effects logistic regression for odds of periodontitis among diabetic patients: Age 1.09 [1.04,1.16], (*p* = 0.003) HbA1c 0.53 [0.27,0.89], (*p* = 0.04) BOP 1.03 [1.00,1.06], (*p* = 0.048) Examining the associations of the parameters with periodontitis only among diabetic patients age, HbA1c, BOP were significantly associated with periodontitis.
Sun et al. [20], 2019	Cohort study	To determine the quantified risk of PD and the influence of emergency visits and hospitalizations in PD development in DM1 patients.	DM1 patients aged <40 years with newly diagnosed DM1 (ICD [International Classification of Diseases]- 9 codes 250.x1 and 250.x3) within the RCIPD (Registry Catastrophic Illnesses Patient Database) from 1 January 1998 to 31 December 2011. The non-DM1 cohort identified subjects without DM1 during (1998–2011)	Individuals with any history of PDs before the index date (ICD-9 Code 523).	4248 DM1 patients [2122 Males/2126 Females] 16992 non-DM1 patients [8504 Males/8488 Females]	<40 years	1998–2011	- Emergency room visits - Annual hospitalizations - Gingivitis - Periodontitis	DM1 patients vs. Non-DM1: (adjusted hazard ratio—aHR (95%CI) - Risk to PD = 1.45 (1.35–1.56); (*p* < 0.001). - Average number of annual Emergency room visit for DM1 ≥ 2 = 13.0 (11.1–15.2) (*p* < 0.001) - Average number of annual hospitalizations for DM1 ≥ 2 = 13.2 (11.5–15.1) (*p* < 0.001) - Risk to develop gingivitis = 1.47 (1.36–1.59) (*p* < 0.001) - Risk to develop periodontitis = 1.66 (1.41–1.96) (*p* < 0.001) The risk of PD was 1.13 (06–1.21) (*p* < 0.001) in patients aged <20 Gingivitis rate DM1 vs. Control (<20 Y): 43.9 vs. 31.4 (20–40 Y): 33.1 vs. 22.9 Periodontitis rate DM1 vs. Control: (<20 Y): 7.75 vs. 5.50 (20–40 Y): 15.2 vs. 8.13 People 20–40 aged have a lower incidence of gingivitis and a higher incidence of periodontitis than those with age < 20 in both case and control groups.
Keles et al. [30], 2020	Case-control study	To compare the Gingival Crevicular Fluid IL-18 and [TNF]-α levels in children with or without DM1 or gingivitis; To investigate whether GCF IL-18 and [TNF]-α are useful markers for gingivitis in patients with DM1.	Age between 8 and 14 years; diagnosed with DM1 by a pediatric endocrinologist at least 12 months prior to the study, with an HbA1c level < 7.5%; having fully erupted caries-free maxillary and mandibular first molars and incisors; do not have any systemic diseases (healthy group).	Having any other known systemic chronic illnesses; HbA1c level > 7.5%; having any destructive periodontal disease or periodontal therapy involved antimicrobial or anti-inflammatory drugs in the past 6 months; having restorative and endodontic therapy requirement; taking immunosuppressive drugs in the past 6 months; taking any medication regularly; having orthodontic treatment and having clinical attachment loss.	44 Systemically healthy children [20 Males/24 females]: Systematic and Periodontally healthy children (H, *n* = 22) Systematic healthy children with Gingivitis (G, N = 22) 44 Children with DM1 [19 Males/25 females]: Periodontally healthy children with DM1(DM1+H, n = 22) DM1 Children with gingivitis (DM1+G, n = 22)	8–14 years	April–June 2019	- HbA1c - PI - PPD - GI - GCF volume - (TNF)-α - IL-18	TNF-α and IL-18 values in DM1 vs. H Children: mean ± SD TNF-α total amount (pg/sample): 3.49 (0.94–5.35) vs. 3.30 (3.62–4.77) IL-18 (pg/sample):0.51 (0.36–0.92) vs. 0.52 (0.42–0.63) (*p* > 0.05). No significant differences in the IL-18 and TNF-α total amounts between the group of DM1 children and the systemically healthy children (*p* > 0.05). H vs. G vs. DM1+H vs. DM1+G: mean ± SD PI 0.30 (0.0–0.60) vs. 1.90 (1.00–2.00) vs. 0.30 (0.0–0.5) vs. 1.89 (0.9–2.79) (*p* < 0.0001) GI (0.54 ± 0.17) vs. (2.07 ± 0.38) vs. (0.51 ± 0.19) vs. (2.24 ± 0.40) (*p* < 0.0001) PPD (mm)—(0.93 ± 0.35) vs. (2.03 ± 0.34) vs. (1.0 ± 0.31) vs. (2.23 ± 0.46) (*p* < 0.0001) TNF-α (pg/μL)—47.93 (24.23–125.30) vs. 15.37 (1.06–33.4) vs. 43.65 (11.66–231.76) vs. 17.39 (8.96–33.40) (*p* < 0.0001) IL-18 (pg/μL): 8.53 (3.76–19.54) vs. 2.05(0.16–3.90) vs. 7.21 (2.79–40.12) vs. 2.49 (1.04–5.45) (*p* < 0.0001) The gingivitis subgroups showed a significantly higher PI, GI, PPD, GCF volume, and TNF-α total amounts than the H subgroups (*p* < 0.0001). - IL-18 concentrations were significantly higher in the periodontally healthy subgroups than in gingivitis subgroups. - TNF-α were positively correlated with PI, GI, PPD, GCF volumes and IL-18 concentration (r = 0.552, *p* = 0.01; r = 0.579, *p* = 0.01; r = 0.534, *p* = 0.01, respectively). - There was a negative correlation between the IL-18 concentration and the TNF-α (−0.524, *p* = 0.01).
Dicembrini et al. [11], 2021	Cross-sectional study	To investigate the prevalence of PD in patients DM1 and its association with glycemic control and glucose variability.	- DM1 patients aged ≥ 18 years and currently treated with multiple daily insulin injections or continuous subcutaneous insulin infusion, who provided their written informed consent and had been continuously using for the last three months the FreeStyle Libre Flash Glucose Monitoring (FGM) system.	Individuals with history of cancer, HIV, bone metabolic disease, history of radiation or immunosuppressive/modulating therapy; those who had taken antibiotics, corticosteroids, or non-steroidal anti-inflammatory drugs in the previous 3 months.	136 DM1 patients [60 Male/76 Female]	19–81 years	12 months	- CAL - PPD - CV - HbA1c	- The prevalence of periodontal disease was 63% (Stage I n = 14; stage II n = 20; stage III n = 43; stage IV n = 9) - A significant correlation was found between mean CAL/CV (r = 0.31, *p* = 0.002), but not HbA1c (r = 0.038 *p* = 0.673). - Mean PPD/CV but not with HbA1c (r = 0.27 and 0.044; *p* = 0.007 and 0.619, respectively). - Multiple linear regression model: assuming the mean CAL as dependent variable, age, CV, and smoking habit resulted significantly associated (r = 0.23, *p* = 0.013; r = 0.33, *p* = 0.001; r = 0.34, *p* < 0.001, respectively). - PPD as a dependent variable showed a significant association with glucose CV and smoking habits only (r = 0.23, *p* = 0.019; r = 0.33, *p* = 0.001, respectively).
Sereti et al. [3], 2021	Case-control study	To evaluate the GCF levels of MMP-8, IL-8 and AGEs in DM1 patients with different glycemic levels and to compare them to healthy controls.	Individuals DM1 aged between 18 and 85 years old, presented at least 10 natural teeth and were diagnosed for DM1 for more than 1 year.	NR	DM1 patients (n = 50 [30 Males/ 20 Females]) Non-diabetic patients (n = 50 [30 Males/ 20 Females])	18–85 years	NR	- HbA1c - MMP-8 - IL-8 - AGEs - PI - GI - BOP	Dental and biochemical parameters between DM1 vs. Non-DM1: mean ± SD GI: 1.1 (0.7) vs. 0.4 (0.4) (*p* < 0.001) BOP: 40.5 (22.2) vs. 29.4 (16.4) (*p* = 0.009) Number of sites PI > 1: 23.9 (27.2) vs. 13.8 (14.5) (*p* = 0.047) IL-8: 220 pg/mL vs. 225 pg/mL, (*p* = 0.433) MMP-8: 32.1 μg/mL vs. 38.3 μg/mL, (*p* = 0.538) AGEs: 3.4 μg/mL vs. 5.8 μg/mL, (*p* = 0.905) - The median GCF levels of MMP-8, IL-8 and AGEs did not differ significantly between groups. - No significant differences were seen in younger (<40 Y) and older (>40 Y) cohorts, in which concern the GCF levels of MMP-8, IL8 and AGEs, between diabetics and controls. - The diabetic group was divided in two sub-groups according to their glycaemic status (HbA1c 6.1–8, and >8%), and no significant differences were observed in GCF between the diabetic subgroups and the controls.
Jensen et al. [26], 2021	Cross-sectional study	To characterize periodontal risk markers (Plaque Index (PI), gingival index (GI), bleeding on probing (BOP) and PPD; To determine the relationship between periodontal risk markers and glycemic control; To determine the relationship between the oral microbiota and both glycemic control and periodontal risk markers.	Individuals aged between 8 and 18 years, who had been previously diagnosed with DM1 by detectable islet cell autoantibodies.	Subjects diagnosed with diabetes other than DM1 or inadequate English language skills to understand the information sheet. Subjects who had an intercurrent fever or infection, diabetic ketosis, or those who were taking antibiotics on the scheduled day of the dental examination were rescheduled.	77 Patients [37 Males/39 Females]	13 ± 2.6 years	February 2018–March 2019	- PI - GI - BOP - PPD - HbA1c	- Median HbA1c of 8.5% (range 5.8–13.3) - 49% had early markers of PD - 1% increase in HbA1c was independently associated with an average increase in BOP OF 25% (*p* = 0.002) an increase in the rate of sites with PPD > 3 mm of 54% (*p* = 0.003) - HbA1c was positively correlated with PI (Rho = 0.34; *p* = 0.002), GI (Rho = 0.30; *p* = 0.009), BOP (Rho = 0.44; *p* = 0.0001), PPD > 3 mm (Rho = 0.21; *p* = 0.06). The worsening of glycemic control is associated with increased severity of early markers of periodontal disease in children and adolescents with DM1.

Legend: AL—Attachment Loss; DM—Diabetes Mellitus; NR—Non referred; OPG—osteoprotegerin; RANKL—receptor activator of nuclear factor kB ligand; BI—Bleeding Index; BOP—Bleeding on probing; CAL—Clinical attachment Loss; DM1—Type 1 Diabetes Mellitus; GMC—Good Metabolic Control; PPD—Periodontal probing Depths; PD—periodontal disease; PMC—Poor Metabolic Control; IL-8—Interleukin-8; BL—Bone Loss; AGEs—Advanced glycation end-products; CP—Chronic Periodontitis; GI—Gingival Index; CRP—C-Reactive Protein; HbA1c—Glycated Haemoglobin; TNF-α—Tumor Necrosis factor α; OHI—Oral Hygiene Index; CPI—Community Periodontal Index; PI—Plaque Index; IL-6—Interleukin 6; CI—Calculus Index; DM2—Type 2 Diabetes Mellitus; HC—healthy controls; REC—Gingival Recession; 95%CI—Confidence Interval; IL-18—Interleukin 18; GCF—Gingival Crevicular Fluid; CV—Glucose Coefficient Variation; MMP-8—Matrix Metalloproteinase 8.

## Data Availability

The data can be accessed by contacting the corresponding author.

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
