# Peer review of "Association between Type 1 Diabetes Mellitus and Periodontal Diseases"

_jcm, 2023, doi:10.3390/jcm12031147_

Round 1

Reviewer 1 Report

Dear Authors,

Your article entitled "Association between Type 1 Diabetes Mellitus and Periodontal Diseases" brings some useful information for the research in this field.

However, I consider that the paper needs some improvements, listed below.

The Introduction section should be more comprehensive and include more information from the literature concerning the relationship between the two studied diseases.

The Materials and methods section:

-given the volume of information existing in the specialized literature on the theme of your article, I consider that too few databases were included in research.

-tables 1, 2 and 3 are not cited in text. A brief text explanation should precede the tables.

The Results section:

-subchapter 3.1 - Figure 1 - a more exact recalculation of the number of articles is necessary, as well as a presentation of the number of articles in each section of the flowchart

-the section should contain a brief presentation in text of the main results.

The Discussion section:

-some of the paragraphs in this section should be moved to the Results section

-please specify the limitations of the study - one of them is represented by the reduced number of searched databases

-this section should also include a presentation of the study bias 

-line 14 of the section: please consider 1.47 instead of 1,47, as well as in other similar cases: lines 28, 30 etc.

The References section should include much more titles, not only 26 (which is the number of articles included in your review).

Briefly, I consider that the paper is not very comprehensive because of the limited number of searched databases and I strongly recommend you to either enlarge the research by including more databases, or to visibly point out this limitation of the study.

Author Response

Thank you very much for your comments/revisions of this article.

We answer all the comments point by point, as indicated bellow:

1- The Introduction section should be more comprehensive and include more information from the literature concerning the relationship between the two studied diseases.

We have included more information from the literature concerning the relationship between type 1 diabetes and periodontal diseases (Hilighted at yellow).

"Type 1 diabetes mellitus is associated.......... glycated haemoglobin" 

2-The Materials and methods section:

2.1-given the volume of information existing in the specialized literature on the theme of your article, I consider that too few databases were included in research.

As suggested, we added more databases that we think that will  enrich the information on the theme of our article(Web of science and Scopus). Thank you for asking this.

2.2-tables 1, 2 and 3 are not cited in text. A brief text explanation should precede the tables.

We have proceed as suggested. We have also include a brief text explanation that precede the tables:

"The degree of quality of the studies on the relational index used and the number of positive responses to the questions in is mostly high 9 articles, although we can also find five studies with moderate evidence and one of low quality." 

3- The Results section:

3.1-subchapter 3.1 - Figure 1 - a more exact recalculation of the number of articles is necessary, as well as a presentation of the number of articles in each section of the flowchart

We have proceed as suggested.

3.2-the section should contain a brief presentation in text of the main results.

In accordance with your suggestion we have added  the phrase: " 

"From each eligible study included in the present systematic review, we collected data about is general characteristics, such as study design and aimed, Inclusion and exclusion criteria, as well as the study population (with sample size and age group or average age), the duration in months or years of the study as follow period and the outcome measures and results (Table 5)."

4-The Discussion section:

4.1-some of the paragraphs in this section should be moved to the Results section

We have improve the discussion issues in the part of discussion and send some paragraphs for previous sections of the article.

4.2-please specify the limitations of the study - one of them is represented by the reduced number of searched databases

We have proceed as suggested. and included our limitation paragraph. please note that we also increase the databases for the search. 

"We acknowledge some limitations in our systematic review, firstly related with the few existing original articles suitable for inclusion (such as randomized-controlled trials), or lack of information that could been used for a quantitative analysis. For these reasons, a metanalysis was not possible. Nevertheless, this study to our knowledge is an original systematic review of the existing data regarding the association between DM1 and PD."

4.3-this section should also include a presentation of the study bias 

We added a sentence of risk of bias/quality of studies:

"When evaluating the quality of the studies, using the Joanna Briggs most of them were classified as high or moderate score quality. The item that presented wrost results were the “strategies to deal with confounding factors state”, with many of them not presenting such cofactors such as tobacco habits. Unfortunately there are no other systematic review to our knowledge to make a comparison to ours results, regarding the quality of the studies (or existence of bias)."

4.4-line 14 of the section: please consider 1.47 instead of 1,47, as well as in other similar cases: lines 28, 30 etc.

We have already changed that.

5- The References section should include much more titles, not only 26 (which is the number of articles included in your review).

Thank you for your suggestion, we have added more 10 references.

6- Briefly, I consider that the paper is not very comprehensive because of the limited number of searched databases and I strongly recommend you to either enlarge the research by including more databases, or to visibly point out this limitation of the study.

We have added two new databases as mentioned above in order to enlarge the research.

Thank you very much for all comments. We think that has improve our article. 

Reviewer 2 Report

Authors have conducted a systematic review on association between type 1 diabetes mellitus and periodontitis. 
The study is well designed and results are documented very well. No further changes are required.

Author Response

Thank you very much for agreeing to review this article.

We are grateful for such positive comments and for considered a well designed study and the results very well documented.

Reviewer 3 Report

The paper, in addition to bringing a scientific importance designed in a systematic review methodology, is outlined in a clear and objective way. In the discussion, perhaps it would be opportune to mention or clarify, based on the literature review very well conducted in the paper, something about the treatment in these cases or the care that would be indicated by the professionals who would be responsible for these patients.

Author Response

Thank you very much for agreeing to review this article.

We are grateful for such positive comments.

In relation to your suggestion, we have put in the last paragraph of the discussion, the importance of communication between the dentist and the endocrinologist in order to control both diseases. Regarding the treatment of periodontal diseases, it is really proven, that improves the glycaemic control of diabetic patients, however, this was not the objective of this study.

Round 2

Reviewer 1 Report

Dear authors,

Thank you for considering my suggestions. What remains to do is some minor English changes and corrections of numbers (such as 1.47 instead of 1,47 etc.).

Author Response

Thank you very much for your comments. As suggested we reviewed the English language and correct the numbers (such as 1.47 instead of 1.47 etc.).

The changes are pink.